# Neutrophils in Inflammatory Diseases: Unraveling the Impact of Their Derived Molecules and Heterogeneity

**DOI:** 10.3390/cells12222621

**Published:** 2023-11-13

**Authors:** Bushra Riaz, Seonghyang Sohn

**Affiliations:** 1Department of Biomedical Science, Ajou University School of Medicine, Suwon 16499, Republic of Korea; bushra0869@ajou.ac.kr; 2Department of Microbiology, Ajou University School of Medicine, Suwon 16499, Republic of Korea

**Keywords:** neutrophil, antimicrobial peptides, neutrophil heterogeneity, autoimmunity, immune responses, inflammatory disease

## Abstract

Inflammatory diseases involve numerous disorders and medical conditions defined by an insufficient level of self-tolerance. These diseases evolve over the course of a multi-step process through which environmental variables play a crucial role in the emergence of aberrant innate and adaptive immunological responses. According to experimental data accumulated over the past decade, neutrophils play a significant role as effector cells in innate immunity. However, neutrophils are also involved in the progression of numerous diseases through participation in the onset and maintenance of immune-mediated dysregulation by releasing neutrophil-derived molecules and forming neutrophil extracellular traps, ultimately causing destruction of tissues. Additionally, neutrophils have a wide variety of functional heterogeneity with adverse effects on inflammatory diseases. However, the complicated role of neutrophil biology and its heterogeneity in inflammatory diseases remains unclear. Moreover, neutrophils are considered an intriguing target of interventional therapies due to their multifaceted role in a number of diseases. Several approaches have been developed to therapeutically target neutrophils, involving strategies to improve neutrophil function, with various compounds and inhibitors currently undergoing clinical trials, although challenges and contradictions in the field persist. This review outlines the current literature on roles of neutrophils, neutrophil-derived molecules, and neutrophil heterogeneity in the pathogenesis of autoimmune and inflammatory diseases with potential future therapeutic strategies.

## 1. Introduction

The ability of organisms to defend themselves against external pathogens and to repair tissue injury brought on by infection is vital for them to survive. Inflammatory processes can contribute to the regulation of the origin, development, and outcomes of autoimmune and autoinflammatory diseases [1]. Moreover, inflammatory responses can damage host tissues and result in organ failure in a wide range of disorders. These inflammatory diseases are characterized by inflammation, which has been identified as the leading cause of death worldwide [2].

Innate immune responses serve as the initial line of defense for the host against pathogens. These responses can help detect and eliminate infected cells as well as coordinate and activate the development of adaptive immunity [3,4]. Neutrophils play a crucial role as effector cells in the innate immune system. Neutrophils are considered a type of polymorphonuclear (PMN) leukocyte. They are emerging as highly specialized cells that are capable of carrying out a number of immune defense-related functions [5]. Neutrophils constitute the most prevalent leukocytes in circulation and possess a short lifespan. If microbial infections are detected, these cells immediately act to trap and eliminate invasive pathogens [6]. In addition, neutrophils interact in complicated, bidirectional ways with certain other immune cells once they reach inflamed tissues, affecting both innate and adaptive immune responses [7].

Furthermore, neutrophils are naturally the first to respond to acute inflammation and aid in its resolution. Neutrophils have additionally been proven to be important in chronic inflammation throughout the past few decades. Neutrophils are persistently recruited towards the site of chronic inflammation. They help drive inflammatory processes by releasing inflammatory factors and several cytokines that can regulate inflammation and activate other types of immune cells [8,9]. Numerous pathological conditions such as cancer, neurological, metabolic, and autoimmune disorders are caused by neutrophil-mediated inflammation [10].

Moreover, neutrophils were previously considered a homogeneous population with conserved phenotypes and distinct roles. Current studies have shown that heterogeneous populations of neutrophils have diverse functional phenotypes, especially in pathological conditions of inflammation [11]. Because neutrophils serve as primary effector cells, the severity and type of inflammatory response after severe injury may be determined by the expression pattern of neutrophil receptors. Neutrophil heterogeneity might serve as a helpful risk assessment tool [12,13].

Many different approaches targeting neutrophils, including strategies that use a number of different agents to improve, hinder, or restore neutrophil activity, have emerged. However, there are still challenges and controversies that remain in the field of neutrophil research. This review emphasizes the pathogenic role of neutrophils and their derived molecules, elucidating their relation to various inflammatory diseases. It also outlines novel insights into phenotypical and functional heterogeneity of neutrophils during inflammatory diseases. In addition, some effective potential therapeutic attempts that specifically target neutrophils are summarized.

## 2. Neutrophil Activation

Neutrophil activation is often a multi-step process that is advantageous for killing pathogens. However, neutrophil activation pathways can also potentially cause tissue damage in autoimmune and inflammatory diseases. Some stimulants, including pathogen chemoattractants (fMLP) and bacterial lipopolysaccharide (LPS), are important for activating neutrophils [14,15]. These stimulants can adhere to neutrophils and activate their receptors to enhance responsiveness of those cells to subsequent stimuli. There are several different kinds of receptors that are expressed on neutrophils, including G-protein-coupled receptors (GPCRs), Fc-receptors, adhesion molecules/receptors such as integrins and selectins/selectin ligands, different cytokine receptors, and innate immune receptors such as C-type lectins and toll-like receptors (TLRs). Activation of such receptors can cause additional reactions such as chemotactic migration or the release of chemokines and cytokines as well as multiple cellular activation and eradication processes such as phagocytosis, generation of reactive oxygen species (ROS), exocytosis of intracellular granules, and the releasing of neutrophil extracellular traps (NETs).

GPCRs like formyl peptide receptors (FPR1, FPR2) have been recognized as bacterial products and mitochondrial peptides via the MAPK/ERK pathway [16], leukotriene B4 (LTB4), platelet-activating factor receptors (PAFRs), complement receptors (C3aR, C5aR1), CXC chemokine receptors (CXCR1, CXCR2, CXCR3, CXCR4), and, to a smaller extent, CC chemokine receptors (CCR1, CCR2, CCR3, CCR5, ACKRs, CCRL2). These GPCRs are capable of enhancing neutrophil responses to further activation. Furthermore, the chemotactic migratory activity of neutrophils is substantially triggered by these receptors [17,18,19]. Several Fc-receptors that are expressed by neutrophils (Fcγ, Fcε, Fcα) have a role in the identification of immunoglobulin (Ig)-opsonized pathogens. They are also involved in immune-mediated inflammatory conditions. Low-affinity Fcγ-receptors are among the most significant Fc-receptors of neutrophils [20,21]. Additionally, neutrophil activation often happens when they bind to extracellular matrix proteins or other cells through integrin or selectin adhesion receptors. Neutrophils can adhere firmly when integrins interact with their specific ligands, such as intercellular adhesion molecules (ICAMs) on endothelial cells. Several kinases, including Src-family kinases, phosphoinositide 3-kinase (PI3K), tyrosine kinase (SYK), and p38 mitogen-activated protein kinase (MAPK), are activated by β2 integrin ligation and selectin–selectin ligand interaction [22,23,24,25]. The activation of these kinases is important for neutrophil functions. Numerous cytokine receptors are also expressed by neutrophils, including tumor necrosis factor (TNF) receptors, granulocyte-colony stimulating factor (G-CSF), and granulocyte macrophage-colony stimulating factor (GM-CSF) receptors, as well as different interleukins (ILs) and interferons (IFN) receptors [22]. These receptors control a variety of neutrophil functions through intercellular communication.

Numerous innate immune receptors, also known as pattern recognition receptors (PRRs), are expressed on neutrophils. They have a direct role in the identification of microbes and tissue injury. According to the homology of protein domains, PRRs can be divided into the following categories: TLRs, retinoic acid-inducible gene-I (RIG-I)-like receptors (RLRs), nucleotide-oligomerization domain (NOD)-like receptors (NLRs) and C-type lectin receptors (CLRs) [22,26,27]. In neutrophils, activation of these receptors is possible through a variety of signal transduction mechanisms. Neutrophil PRRs are important regulators of host immunological responses.

Consequently, these neutrophils’ stimulated receptors can detect microbes as well as an inflammatory environment. Understanding these neutrophil receptors and related signaling pathways that regulate neutrophil function is necessary for the development of therapies that can prevent host tissues from being harmed by neutrophils.

## 3. Structure and Functions of Neutrophils

Neutrophils are composed of different granule types: (1) primary granules, commonly referred to as azurophilic granules; (2) secondary granules, also called specific granules; (3) tertiary granules, also known as gelatinase granules; and (4) secretory granules. Primary granules contain most mediators, including elastase, MPO, proteinase 3, Cat-G, azurocidin, and defensins. Secondary granules are the storage site of lactoferrin, cathelicidin, neutrophil gelatinase-associated lipocalin (NGAL), and collagenase. Tertiary granules comprise matrix metalloproteases (MMPs). Secretory granules contain plasma proteins and cationic antimicrobial protein 37 (CAM37). Activated neutrophils can also release other effector molecules such as reactive oxygen species (ROS), LTB4, calprotectin, peptidyl arginine deiminase (PAD), NETs, and various types of cytokines and chemokines, as shown in Figure 1.

Neutrophils are multifaceted cells with a wide range of distinct functions. Appropriate neutrophil recruitment is required for modulation and resolution of inflammation, tissue repair, wound healing, elimination of microorganisms, and restoration of homeostasis, as depicted in Figure 2. These various roles that neutrophils play will be discussed below.

### 3.1. Chemotaxis

Neutrophils can identify inflammatory signs and move in the direction of infected sites [28]. For this efficient response, they can recognize extracellular chemical gradients and migrate toward higher concentrations through a process known as chemotaxis. Chemoattractants are a set of molecular guidance cues with diverse chemical compositions. They are responsible for orchestrating this targeted neutrophil recruitment. In humans, these chemoattractant molecules can be classified into four molecular families: formyl peptides, chemokines, anaphylatoxins, and chemotactic lipids [29]. Neutrophil chemoattractants work by interacting with heptahelical GPCRs expressed on cell surfaces. The Rho family of GTPases play a major role in controlling chemotactic response. Moreover, emerging data indicate that the atypical chemoattractant receptor (ACKR), another receptor that does not bind to G proteins, might also play a significant role in regulating the migratory ability and functional responses of neutrophils. The expression of chemoattractant receptors is reliant on the degree of neutrophil maturation state and activation, with a crucial modulatory function for an inflammatory condition [17].

### 3.2. Killing of Microbes

Neutrophils can phagocytose microorganisms at sites of infection. Throughout this process, both primary and secondary granules combine well with phagosomes and produce proteins and antimicrobial peptides such as myeloperoxidase (MPO), neutrophil elastase (NE), cathepsin G (Cat-G), cathelicidin, alpha-defensins, and many others. During the same periods, ROS are generated via nicotinamide adenine dinucleotide-phosphate (NADPH). NETs are produced by activated neutrophils in response to specific stimuli. NETs resemble a net-like network made up of cell-free DNA, neutrophil granule proteins, and histones. NETs are recognized as a pathogen control strategy identified only a few years ago [30]. All of these microbicidal molecules are secreted by neutrophils to create a highly dangerous environment that seems to be necessary for effective microbial killing and destruction. Several of these microbicidal molecules can aid in the development of infection by being cytotoxic to host tissues [31,32]. Thus, it is expected that the host uses a variety of mechanisms to restrict or stop them from harming host cells and causing undesired inflammation.

### 3.3. Resolution of Inflammation

Neutrophils have a short lifespan. They are limited by apoptosis. Apoptotic neutrophils are functionally inactive due to a programmed shutdown and disabling of their signaling pathways. However, they can also retain cell-surface receptors that allow them to be recognized and phagocytosed by macrophages and other phagocytic cells. By removing apoptotic neutrophils from inflammatory areas, it is possible to avoid tissue damage that might otherwise occur due to the discharge of cytotoxic compounds into surrounding tissues that might have died due to necrosis. The efficient death of neutrophils and the secure evacuation of apoptotic neutrophils through phagocytic cells are crucial for the reduction in inflammation [9,33]. Under inflammatory conditions, disruption of neutrophil apoptosis can result in prolonged survival of neutrophils in damaged tissue and prolong the secretion of neutrophil-derived immunomodulatory cytokines, cytotoxic chemicals, and chemokines, which may lead to prolonged inflammation [34,35]. Failure in the removal of apoptotic neutrophils could also result in the generation of autoantibodies, because these cells express autoantigens upon their surface [36].

### 3.4. Neutrophil Network with Other Immune Cells

Neutrophils not only participate in the eradication of microbes, but also promote immune reactions to intracellular pathogens through intricate interactions with other immune cells. For instance, neutrophils can release chemokines (such as CCL3, CCL4, CCL5, and CCL20) and alarmins, including α-defensins, cathelicidins, and high-mobility group box-1 (HMGB1) proteins that are chemotactic for dendritic cells (DCs). These chemokines and alarmins are required for efficient DC recruitment to infected sites. NETs can promote plasmacytoid (p) DCs to release inflammatory cytokines [37]. Similarly, interactions between macrophages and neutrophils play a crucial role in both the beginning and resolving stages of an inflammatory reaction. Tissue-resident macrophages can secrete some chemoattractants like CXCL1, CXCL2, CCL2, and IL-1α that are required for the migration of activated neutrophils toward the inflammatory site. This process can increase the lifespan of neutrophils by secreting G-CSF, GM-CSF, and TNF-α [38]. When neutrophils arrive at the site of inflammation, they can activate the immune system by recruiting monocytes and releasing proteins such as LL-37, proteinase 3 (PR3), azurocidin, defensins, and Cat-G [39,40]. Defensins, azurocidin, and other antimicrobial peptides can boost antimicrobial activities of macrophages by enhancing their capacity to phagocytose and generate cytokines (TNF-α and IFN-γ) [41,42].

Cytokines BAFF (B-cell activating factor) and APRIL (a proliferation-inducing ligand) are generated by neutrophils in great quantities. These cytokines are essential for B-cell survival, development, and differentiation. They are elevated in inflammatory diseases [43,44]. Furthermore, the function of several T-cell subsets can be positively or negatively modulated by neutrophils. In both humans and animals, activated neutrophils encourage T-cell activation, multiplication, and differentiation into effector CD8+ T-cells, and T-helper cell subsets (Th1, Th17) could promote adaptive responses at the inflammatory site [45,46,47]. According to previous studies, granule peptides of neutrophils such as cathelicidin (mCRAMP in mice, LL-37 in humans) can exert immunomodulatory actions on T cells and modulate Th1 and Th17 differentiation [48]. At the site of inflammation, neutrophils can also stimulate natural killer (NK) cells. Cat-G, defensins, elastase, and lactoferrin (LTF) are implicated in the augmentation of cytotoxic activity of human NK cells. The survival rate of human neutrophils can be improved by NK-derived substances including GM-CSF and IFN-γ [49]. These findings suggest that human neutrophils have a variety of mechanisms through which they can control the activity of NK cells.

Crosstalk between neutrophils and other immune cells as well as significant chemical signals can affect the development and remission of inflammation. However, further research on associations of neutrophils with other immune cells in relation to inflammatory diseases is required.

## 4. Neutrophils in Infection

Neutrophils are essential mediators that can serve in the initial defense against invasive pathogens like bacteria and viruses. Next, we will explain the effector role of neutrophils in bacterial and viral infections.

### 4.1. Bacterial Infection

Neutrophils are required as a key component of the innate response to bacterial infection. Upon bacterial infection, neutrophils will exit the bloodstream and move to the inflammatory area to fight against bacterial infections. When an infection is caused by *Listeria monocytogenes*, a Gram-positive intracellular pathogen, neutrophils can move from the bone marrow to the infectious site. They employ unique bacterial-sensing mechanisms at this site that can result in phagocytosis and generation of bactericidal substances [50]. Liu et al. [51] have investigated formyl peptide receptors related to chemoattractant GPCRs which are crucial for the quick migration of neutrophils in *Listeria*-infected livers of mice for successful clearance of infectious microbes. LTB4, another chemoattractant for neutrophils, is essential for neutrophil colonization. A recent publication has shown that preincubation of human neutrophils with the Gram-negative intracellular bacteria *Salmonella typhimurium* can promote neutrophil colonization [52]. This bacterium has been used to stimulate LTB4 production caused by a bacterial chemoattractant fMLP, which is important for the eradication of pathogens instantaneously. However, certain bacterial infections also generate molecules that can inhibit neutrophil recruitment. Extracellular bacteria like *Streptococcus pyogenes* (*S. pyogenes*) and *Streptococcus pneumoniae* (*S. pneumoniae*) use a significantly different strategy to prevent neutrophil recruitment. For example, streptolysin is an effective cytolytic toxin produced by *S. pyogenes*. This streptolysin is essential for inhibiting neutrophil recruitment in the initial stages of *S. pyogenes* infection, as observed in studies using zebrafish [53]. Similarly, zinc metalloproteinase produced by *S. pneumoniae* can cleave P-selectin glycoprotein 1 (PSGL-1), preventing neutrophil extravasation in its first stages [54]. Moreover, *Staphylococcus aureus* is an intracellular bacterium with chemotaxis inhibitory protein that has been employed to prevent neutrophil activation [53,55,56,57].

Additionally, phagocytosis, a mechanism that occurs in neutrophils, allows them to eliminate bacterial infections. There are multiple examples of bacteria that can secrete different compounds to either increase or decrease the phagocytic activity of neutrophils. For instance, *Neisseria gonorrhoeae* can alter mitochondrial depolarization and caspase activation to regulate phagocytosis in human neutrophils [58]. In contrast, *S. pneumoniae* capsules can reduce bacterial opsonization and inhibit effective recognition by complement receptors, Fcγ receptors, and nonopsonic receptors. Due to this inhibition, neutrophils are unable to phagocytose the bacterium, which allows *S. pneumoniae* to cause diseases [59]. Other bacteria including *Mycobacterium tuberculosis*, *Neisseria meningitidis*, *Haemophilus influenzae*, *Pseudomonas aeruginosa*, and *Escherichia coli* can also inhibit the phagocytic activity of neutrophils through different mechanisms [57,60,61,62].

Moreover, neutrophil apoptosis is a pro-resolution process that can reduce the severity of tissue damage and inflammation. However, accelerating or delaying neutrophil apoptosis might have several negative consequences. Some bacterial pathogens such as *Pseudomonas aeruginosa* can release a pigment known as pyocyanin and exotoxin A that can cause apoptosis of neutrophils [63]. Other examples of bacteria that can persuade apoptosis of neutrophils following phagocytosis include *Salmonella typhimurium*, *Escherichia coli*, and *Staphylococcus aureus* [64,65,66,67]. Numerous bacteria have been proven to cause neutrophil apoptosis, although fewer bacteria have been found to be able to prevent this death process. For example, *Chlamydia psittaci*, *Francisella tularensis*, and *Anaplasma phagocytophilum* are other intracellular bacterial pathogens that can delay neutrophil apoptosis via signaling pathways and anti-apoptotic proteins.

NETs are produced by neutrophils through a process known as NETosis, which traps a variety of bacteria. The immunological response to bacterial infections is highly dependent on NETs. NETs can inhibit the growth of bacteria including *Shigella flexneri*, *Escherichia coli*, *Pseudomonas aeruginosa*, *Salmonella typhimurium*, *Shigella sonnei*, *Klebsiella pneumoniae*, *Salmonella enteritidis*, *Staphylococcus albus*, *Pseudomonas aeruginosa*, *Propionibacterium*, and *Staphylococcus aureus* and kill them [68,69].

Overall, the communication between neutrophils and bacteria is dynamic and complex. These are only a few examples. There are undoubtedly many more bacteria that can affect neutrophils. The role of neutrophils during various other bacterial infections is still largely unexplored. Understanding the role of neutrophils in the defense against bacterial infection can be extremely helpful in the development of new therapies for bacterial infection.

### 4.2. Viral Infection

Neutrophils have a variety of functions in severe viral infections. They can limit viral replication and transmission by phagocytosis, respiratory burst, degranulation, cytokine production, antimicrobial peptides, formation of NETs, and activation of the adaptive response. However, excessive activation of neutrophils can harm the tissue with negative effects. Many viruses such as herpes simplex virus (HSV) [70], respiratory syncytial virus (RSV) [71], influenza A virus (IAV) [72], and human immunodeficiency virus (HIV) [73] can activate neutrophils via PRRs to release proinflammatory cytokines, chemokines, ROS, and granular enzymes. Furthermore, Cloke et al. [74] have revealed that certain neutrophil phenotypes like low-density granulocytes (LDGs) are associated with certain viral infections. In HIV-positive patients, LDGs are presumably activated neutrophils that are primed to degranulate. Contrarily, hepatitis C virus (HCV) can decrease neutrophil phagocytosis in both cirrhotic and non-cirrhotic individuals, indicating that neutrophil dysfunction is associated with HCV replication [75]. Research on mice infected with IAV has revealed that blocking C5a can reduce neutrophil recruitment to lungs and tissue damage [76].

It has been demonstrated that many viruses can directly or indirectly promote or prevent neutrophil apoptosis. For instance, the influenza virus can accelerate neutrophil apoptosis. Studies have demonstrated that exposure to the influenza virus might cause neutrophils to upregulate pro-apoptotic factors like Fas and TNF-related apoptosis-inducing ligands (TRAIL). As a result, caspase-8 and caspase-3 are activated, starting the apoptotic cascade to help eliminate infected cells [77]. Several viruses such as HCV, HIV, and Simian immunodeficiency virus (SIV) have been found to increase neutrophil apoptosis. These viruses can cause neutrophil apoptosis by a variety of complex mechanisms/pathways, including the secretion of ROS and cytokines [78,79,80]. When these viruses cause enhanced apoptosis of neutrophils, they can induce neutropenia, potentially weakening the immune response to infection [81,82]. In contrast, delaying neutrophil apoptosis has the potential to worsen tissue damage and accelerate viral clearance. Therefore, the host may benefit from suppression of neutrophil apoptosis. For instance, it has been revealed that human cytomegalovirus (HCMV) can inhibit neutrophil apoptosis and cause production of a highly bioactive secretome (TNF-α, IL-6, IL-8, MIP-1α, IL-13, and IL-10) that can promote neutrophil survival and trigger the chemotaxis of monocytes and their differentiation into permissive, anti-inflammatory phenotypes [83]. In addition, one research paper has demonstrated that RSV can also extend the longevity of human neutrophils by preventing or delaying apoptosis [84].

Likewise, NETs can trap and eliminate viral infections. Certain viruses, including influenza A, HIV-1, and RSV, can cause the development of NETs. By generating ROS species and activating TLRs 4, 7, or 8, these viruses can trigger NETosis, a process by which NETs can trap and destroy viruses [85,86,87,88]. Nevertheless, acute viral infections such as those brought by dengue virus (DV) and coronavirus-2 are known to generate dysregulated NETs. Dysregulated NET formation has been demonstrated to be a measure of disease severity. It plays a part in the development of infection [89].

In summary, viruses can affect neutrophil activities in a variety of ways, which can affect immune responses and aid in the emergence of viral infections. Awareness of complex interactions among neutrophils and viruses can offer useful insights for the development of efficient therapies against viral diseases.

## 5. Role of Neutrophils in Inflammatory Diseases

Inflammatory diseases are defined by chronic or persistent inflammatory responses that can cause tissue injury and malfunction. Neutrophils, their derived molecules, and neutrophil heterogeneity all have an effective role in the emergence and perpetuation of inflammatory diseases, including multiple sclerosis (MS), inflammatory bowel disease (IBD), Behçet’s disease (BD), atopic dermatitis (AD), rheumatoid arthritis (RA), and systemic lupus erythematosus (SLE), as shown in Figure 3 [32,90]. Our expanding knowledge of neutrophil function in different inflammatory diseases might have a considerable impact on the development of targeted treatments for inflammatory diseases.

### 5.1. Multiple Sclerosis

MS is an immune-mediated, demyelinating, chronic inflammatory, and neurodegenerative disease of the central nervous system (CNS). It has an unknown etiology. Although the majority of inflammatory cells related to MS are macrophages and T lymphocytes that aggregate inside perivascular regions and brain parenchyma, evidence shows that neutrophils also have a negative impact on the development of MS [91,92]. Recent research has shown that in the early phase of the disease, levels of neutrophils and other leukocytes are much higher in MS patients [93]. During MS, neutrophils can infiltrate into the CNS and lead to tissue destruction and inflammation, which are hallmarks of the disease [94]. Moreover, in the latest genome-wide association study (GWAS) of MS, neutrophil cytosolic factor-4 (NCF-4) gene, encoding one of the subunits of the nicotinamide-adenine dinucleotide phosphate (NADP) complex in neutrophils, was discovered as a genetic factor susceptible to MS [95]. Moreover, neutrophils can promote immune-mediated demyelination in experimental autoimmune encephalomyelitis (EAE), a murine MS model [96]. Collectively, neutrophils exhibit a wide range of effector actions that facilitate the disease pathogenesis.

#### 5.1.1. Neutrophil-Derived Molecules

The pathogenesis of MS involves the production of toxic and immunoregulatory chemicals by neutrophils. For instance, MPO can promote the activation and accumulation of neutrophils in the CNS. MPO activity is high in MS patients [97]. Cortical demyelination has been related to significantly increased MPO activity in a homogenate sample of the MS cortex [98]. Antineutrophil cytoplasmic antibodies (ANCAs) are autoantibodies that can specifically target antigens within cytoplasmic granules of neutrophils [99]. Moreover, a higher level of NE is also secreted by neutrophils of MS patients. NE can trigger the degradation of tissue by cleaving thrombomodulin and damaging tissue proteins, leading to axonal loss in acute and chronic brain lesions [96]. Cat-G has an impact on pathogenic mechanisms of MS by degrading immunodominant myelin basic protein (MBP) epitope, removing its binding to MHC class II, and abrogates MBP-specific T cell response [100,101]. Cathelicidin has a role in a mouse model of MS [102]. Cathelicidin can promote Th17 cell plasticity and differentiation, which results in the production of IFN-γ producing cells in the CNS that might cause inflammation in an EAE model [103]. Higher NGAL production has been observed in the cerebrospinal fluid (CSF) of progressive MS patients [104]. On the contrary, a recent research study has suggested that NGAL might be a protective molecule in the formation of MS lesions in mouse models [105]. Neutrophils can also release other important mediators, including MMPs and ROS. Levels of MMPs and ROS are higher in MS patients as compared to a healthy control group. These factors all could have a role in the disruption of the blood-brain barrier (BBB), brain extracellular matrix (ECM), and brain tissues. They might promote the development of neuroinflammation in MS [106,107,108]. According to previous research, LTB4 levels are noticeably higher in CSF of MS patients than in controls [109]. These findings have been confirmed in an experimental model of EAE [110]. LTB4 might play a role through interactions with its receptors BLT1. This interaction may result in recruitment and activation of immunocompetent cells across inflammatory lesions in addition to an elevation of autoimmune responses [109,110,111]. When compared to healthy controls, MS patients have greater serum calprotectin levels, which are linked to disease activity [112]. Neutrophil-derived cytokines such as TNF, IL-1β, IL-6, and IL-17 are related to higher neuroinflammation in MS patients [113,114,115]. Neutrophils have a powerful chemoattractant known as CXCL1. Increased expression levels of CXCL1 in the CNS can result in enhanced recruitment of neutrophils towards the CNS. Thus, neutrophil recruitment can promote demyelination in the EAE model [116]. In addition, increased levels of other neutrophil-activating chemokines (CXCL1, CXCL5, and CXCL8), which are linked to the development of inflammatory lesions, have been seen in blood samples of MS patients and the EAE model [96]. Furthermore, higher expression levels of neutrophil receptors such as FPR1, CXCR1, TLRs 2 and 4 in MS patients may indicate the potential involvement of neutrophils in the etiology of this disease.

In MS patients, PAD2 and PAD4 levels are high. These enzymes cause MBP to be citrullinated more frequently, which degrades myelin. Citrullinated MBP is a significant player in the pathophysiology of MS [117]. The development of NETs also depends on PAD4 [118]. NETs are more prevalent in individuals with relapsing remitting MS than in individuals with HC or primary progressive MS. Further research has revealed that MPO-DNA complexes are significantly greater in male patients who typically have a worse prognosis than in female patients. The same study has raised the possibility that NETs could harm nearby neurons and other CNS cells by having a cytotoxic effect on the BBB [119]. However, evidence is still required to further show the role of NETs in BBB breakdown in MS.

#### 5.1.2. Neutrophil Heterogeneity

LDGs have been found in significantly higher concentrations in MS patients [120]. According to a previous study, MS patients have considerably more CD16^high^ cells in their LDG fraction, which is a hallmark of mature neutrophils [121]. Furthermore, the neutrophil-to-lymphocyte ratio (NLR) has been suggested as a biomarker for MS disease severity and to predict possible risk of relapse [122,123]. Granulocytic myeloid-derived suppressor cells (G-MDSCs) are significantly more prevalent in active MS patients compared to those in remission with a strong inhibitory effect on the activation and proliferation of autologous T cells [124]. Distinct subpopulations or mature phases of neutrophils can have different impacts in MS.

Overall, the activity of neutrophils in MS is complicated and poorly understood. Ongoing research is actively exploring the role of neutrophils, their specific mediators, and subsets in the progression of MS. Their contribution to inflammation and tissue destruction in the CNS emphasizes the significance of developing novel treatment approaches that can specifically target these cells to treat MS.

### 5.2. Inflammatory Bowel Disease

The most common diseases among BD are Crohn’s disease (CD) and ulcerative colitis (UC). They are mainly characterized by severe inflammation of the gastrointestinal (GI) tract. The actual origins of IBD are unknown. Studies have identified pathogenic immune cell networks and abnormal immune cell trafficking as crucial drivers of tissue damage and mucosal inflammation in IBD [125]. Among various immune cells, the migrating activity of neutrophils toward the colon mucosa is a specific feature of IBD. Neutrophils are thought to perform dual functions in IBD. Firstly, they move toward the intestinal lining to assist in the defense against dangerous bacteria and other infectious agents. Secondly, neutrophils are constantly accumulating or becoming active inside the intestinal mucosa. They can exceed the number of scavenger cells. Prolonged and excessive activation of neutrophils might result in chronic inflammatory processes in IBD [126,127]. However, the exact role of neutrophils in IBD is unclear yet and appears to vary depending on the experimental conditions.

#### 5.2.1. Neutrophil-Derived Molecules

At the inflamed site, neutrophils produce a range of enzymes, chemicals, and inflammatory compounds to kill pathogens. This could also harm surrounding tissues. For instance, IBD patients have increased serum and fecal MPO levels [128]. MPO activity can perpetuate inflammation, which further leads to destruction of host tissues [128,129]. Furthermore, NE has higher concentrations inside the intestinal mucosa of UC patients than CD patients and healthy controls [130,131]. NE may hinder mucosal healing by reducing the proliferation of epithelial cells. Cat-G is upregulated in UC patients, which implies that only UC patients, not normal individuals, can discharge Cat-G from the colon wall to the lumen. Cat-G can also activate the PAR4 receptor. Overexpression of Cat-G in UC patients is related to higher PAR4 expression. We can assume that Cat-G and PAR4 play key roles in the initiation and/or progression of relapses in UC. Additionally, Cat-G has the ability to raise levels of angiotensin II in inflamed areas. This in turn causes death of epithelial cells and disrupts barrier functions of the epithelium [132,133,134]. α-defensins is expressed in epithelial cells of the mucosa of active IBD patients. Patients with UC have elevated concentrations of plasma α-defensins [135]. Fecal LTF level is considerably greater in an active state of IBD than in an inactive state of IBD. By using this protein, it is reliable to distinguish between inflammatory and non-inflammatory IBD [136]. Children with CD and UC have higher serum levels of cathelicidin [137]. Patients with IBD have higher colonic expression levels of cathelicidin in their intestinal mucosa [138,139]. It has been hypothesized that increased cathelicidin production in the inflamed mucosa of IBD might promote antibacterial and anti-LPS activities [140]. It can protect tissues against microbial invasions and excessive inflammatory responses. Another possible biomarker for IBD is fecal NGAL. Patients with active CD and UC have considerably higher fecal NGAL levels than healthy controls and inactive patients of UC and CD [141]. In addition to being a significant source of MMP-9, neutrophils can contribute to epithelial injury in UC patients [142]. In injured intestines of IBD patients as well as in animals with DSS-induced colitis, MMP-8 and MMP-9 levels are elevated [143]. Cells from inflamed IBD epithelium show enhanced proteolytic activity of MMPs [144,145,146]. When neutrophils are activated, they produce excessive ROS, which impairs intestinal homeostasis in IBD [147]. According to previous studies, ROS can increase expression levels of genes related to both adaptive and innate immune responses in the GI tract [148,149]. It has been reported that LTB4 is upregulated in patients with IBD [150]. Neutrophil accumulations are found with LTB4 at sites of cell death in lesional tissues [151,152]. Fecal calprotectin is an indicator of disease activity, especially in UC patients [153]. Moreover, there are other factors that can influence neutrophil activation and infiltration into IBD-related inflamed mucosa. For example, elevated expression of chemokines (CXCL-1, -8, and -10) and cytokines (IL-6, IL-8, IL-1β, TNF-α, G-CSF, GM-CSF, and IL-17) released by neutrophils and other immune cells (intestinal epithelial cells, macrophages) throughout intestinal inflammation is a significant factor in the infiltration, migration, and activation of neutrophils into mucosal surfaces [154,155,156,157,158].

Compared to CD patients and the control group, the inflamed mucosa of UC patients shows a considerably higher level of PAD4 expression [159]. NETs are essential as a defense mechanism when high quantities of viruses, bacteria, and fungi are present in the intestinal mucosa interphase. On the contrary, NETs might be involved in IBD-related problems and intestinal inflammation. Moreover, NET release is seen in UC [160], supporting the idea that NETs play a role in maintaining mucosal inflammation throughout this disease [160,161].

#### 5.2.2. Neutrophil Heterogeneity

NLR might be a promising IBD biomarker. NLR values are significantly higher in those with active IBD than in those with inactive IBD and healthy controls [162]. *Helicobacter hepaticus* (Hh)-induced colitis is a model of IBD in which G-MDSCs promote inflammation by suppressing T-cell activity and modulating innate immune responses. G-MDSCs are accumulated in the spleen and colon samples of the Hh-infected IBD model. Therefore, it is considered that G-MDSCs have a pro-inflammatory role in colitis [163]. CD177+ neutrophils are responsible for negatively regulating the development of IBD. CD177+ cells produce more IL-22 with increased bactericidal activity compared with CD177− subset, suggesting a potential protective function in IBD. Accordingly, the intestinal barrier is impaired in colitis mice lacking CD177+ neutrophils, and the development of colitis is accelerated. Upregulation of CD177+ neutrophils could be beneficial for the management of IBD [164].

### 5.3. Behçet’s Disease

BD is known as an inflammatory disease with numerous manifestations. It can be recognized by the involvement of the vascular, articular, neurologic, and gastrointestinal systems as well as ocular, skin, genital, and oral ulcers [165]. Although the exact etiology of BD is unidentified, immunological irregularities play significant roles in the pathogenesis. It is well known that the pathogenesis of BD is associated with increased neutrophil influx and disturbed self-tolerance [166,167]. Neutrophils usually participate in perivascular infiltration in lesions and show significant intrinsic activity in BD patients. Chemotaxis and phagocytosis can both be increased by hyperactive neutrophils [168,169]. These heightened neutrophil functions contribute to tissue damage and immunological alterations observed in the disease. Patients with active BD have higher levels of neutrophil activation markers including CD64 [170]. Furthermore, neutrophil inflammation is a major mediator of thrombosis in BD. There are also reports of circulating neutrophil impairments in BD due to activation-induced cell death (AICD). A histological study of BD lesions has revealed arterial and venous infiltrates of neutrophils, indicating that neutrophils are specifically implicated in BD lesions [171]. It has been suggested that BD can be categorized as neutrophilic vasculitis [172,173]. Overall, neutrophils are mediators of inflammation that occur in BD patients.

#### 5.3.1. Neutrophil-Derived Molecules

Neutrophil-mediated compounds can aid in the inflammation and destruction of tissues in BD. Le Joncour et al. [174] have stated that MPO-DNA complexes are substantially more prevalent in the serum of BD patients. In addition, the generation of thrombin in BD plasma is markedly elevated. Such elevation is associated with concentrations of MPO-DNA complexes. Furthermore, there are noticeably higher amounts of NE in plasma [175] and saliva [176] of active BD compared to those in inactive and healthy controls. Studies have shown that ANCAs are correlated with vascular involvement in BD patients [177,178]. Regarding defensins, it is known that neutrophil-derived α-defensins have anti-inflammatory action by suppressing macrophage mRNA [179]. α-defensins have been found in the saliva of active BD patients with oral ulcers [180]. In addition, calprotectin and LTF as two fecal biomarkers can be used to identify intestinal inflammation. BD patients with intestinal lesions have significantly higher levels of fecal calprotectin and fecal LTF than patients without lesions [181]. LL-37 is known as an inflammatory and anti-inflammatory peptide [182,183]. Mumcu et al. [184] have shown that salivary levels of LL-37 are positively correlated with oral ulcers in BD patients. LL-37 combination with healthy plasma extracellular vesicles can upregulate proinflammatory cytokines such as IL-1 beta and IL-6. In BD, LL-37 circulates, binds to plasma extracellular vesicles, and causes severe BD symptoms [185]. NGAL has inflammatory properties. It might be employed as a diagnostic marker in ocular BD. NGAL values are much higher in ocular active BD than in healthy controls [186]. Higher concentrations of MMP-9 in skin tissues of BD patients are evaluated, which may promote the severity of BD. Sera of vasculo-BD patients also contain a higher level of MMP-9 [187,188]. Likewise, ROS anomalies caused by neutrophils could be crucial in BD. Increased production of ROS is associated with fibrinogen carbonylation impacting the structure and function of fibrinogen. Moreover, ROS exhibited a marked effect on fibrinogen polymerization, clotting parameters, and fibrin susceptibility to plasmin-induced lysis. There is evidence that neutrophil ROS can increase fibrinogen oxidation, alter clotting architecture, and enhance thrombus progression in BD [189,190]. In BD, the inflammatory activity of neutrophils is possibly regulated by the secretion of various cytokines (IL-17, TNF-α, IL-1β, IFN-I), chemokines, and chemokine receptors (CXCL8, CCR2, CXCR2, CCR1) [168,191,192,193,194,195].

Neutrophils in the blood of BD patients are more likely to express greater amounts of PAD4 than in normal volunteers. Notably, higher levels of NETs and their associated markers have been reported in patients with BD who have vascular complications. NETs can increase thrombosis [166,174]. Neutrophils from active BD patients can release more NETs in response to CD40L than neutrophils from inactive BD patients [196].

#### 5.3.2. Neutrophil Heterogeneity

NLR might serve BD activity because it is substantially higher in active BD patients than in healthy controls and inactive BD patients [197]. Enhanced low-density neutrophils (LDN) levels might also play a role in the etiology of BD and inflammatory responses of BD patients [198]. Neutrophil heterogeneity has not yet been thoroughly investigated in BD. Therefore, various functions of neutrophil heterogeneity in BD remain unclear.

### 5.4. Atopic Dermatitis

AD is a chronic inflammatory skin disease that causes extremely itchy redness and inflamed skin. The most prevalent feature of atopic diseases is immune globulin E (IgE)-mediated allergic reactions related to environmental allergens. AD is considered a heterogeneous disorder with a spectrum of morphology, dispersion, and disease progression. The pathophysiology is complicated, and numerous cell types are involved, including immune cells, skin cells, and neuronal cells that monitor and control immune responses [199,200]. Neutrophils are the initial immune cells that infiltrate AD skin. Moreover, neutrophils can regulate early skin hyperinnervation and upregulate the expression of activity-induced genes and itch-signaling molecules in nerve cells [201,202]. These findings imply that neutrophils might influence AD onset and progression via a variety of mechanisms.

#### 5.4.1. Neutrophil-Derived Molecules

According to recent studies, both infants and adults with AD may experience clinically severe allergic contact dermatitis (ACD) issues. ACD seems to be more likely to occur in AD patients [203]. Contact hypersensitivity (CHS) is known as one of the most frequently used animal models of ACD [204]. The CHS mouse model requires two phases: induction and elicitation. MPO serves dual roles in the development of CHS pathogenesis. During the induction phase, MPO enhances the synthesis of IL-1β under the skin and the migratory activity of DC, which aids in effector T-cell priming. At the elicitation stage, MPO promotes vascular permeability, leading to inflammatory responses. Moreover, blood samples from children with AD have higher levels of MPO and MMPs than samples from adults [205]. Elevation of elastase activity has been shown in peripheral blood neutrophils of AD patients. These elevations can harm and impair skin barrier function [206]. This indicates that the activity of neutrophils with elastase secretion is held in the acute stage of AD. Skin samples from AD patients have much greater Cat-G levels than those from normal controls and are associated with disease severity [101,207]. Proteinase-activated receptor-2 (PAR-2), a protein related to itching, can be activated by Cat-S. PAR-2 aids in the sudden development of skin disorders comparable to chronic AD in Cat-S overexpressing transgenic (TG) mice [208]. Plasma levels of α-defensin are increased after aggravation of AD. These plasma levels of α-defensin are positively associated with AD clinical outcomes, IgE levels, and serum IL-8 levels but inversely associated with serum IL-10 levels [209]. Patients with AD are susceptible to a chronic inflammatory eye condition known as atopic keratoconjunctivitis (AKC). According to Fujishima et al. [210], LTF may regulate some atopic immunological processes in patients with AKC. LL-37 also has a vital role in AD. An increased expression level of LL-37 in lesioned skin of AD patients as compared to non-lesioned skin indicates a potential involvement of LL-37 in AD. It might be related to re-epithelialization processes [211,212]. These studies have suggested that the skin of AD patients may exhibit dysregulation of α-defensins, lactoferrin, and LL-37 [213,214]. Allergic patients, especially those with AD along with systemic inflammation, exhibit elevated NGAL levels in their blood [215]. Furthermore, the pathogenesis of AD and other cutaneous disorders is influenced by the high production of ROS. Higher ROS production is associated with chronic inflammatory activation in AD [216]. In addition, LTB4 is critical for the emergence of skin allergic inflammation in a mouse model having AD features. In injured skin, LTB4 and its receptor BLT1 are critical for the accumulation of neutrophils and neutrophil-dependent recruiting of effector T cells. These findings prove that LTB4-BLT1 interaction is essential for skin allergic inflammation in an AD mouse model [217]. Neutrophils are essential for the induction and activation of CXCL10, which is a ligand of the CXCR3 receptor. CXCL10/CXCR3 signaling shows a particular link between nerve cells and infiltrating neutrophils that can promote itch behaviors in AD mice [202]. In AD patients, higher expression levels of CXCR2 and its ligands have been observed. They can drive neutrophil recruitment toward skin tissues [218,219,220]. Choy et al. [221] have compared transcriptomic profiles of healthy and AD skin samples and revealed that neutrophil chemoattractants (such as GM-CSF and CXCL8) and neutrophil infiltration into the dermis are dramatically increased in AD skin compared to healthy controls.

NETs have been found to have an indirect relationship with AD [222,223]. For instance, *Staphylococcus aureus* colonization is increased on both inflamed and non-inflamed skin of AD patients where it can further promote skin inflammation [222]. NET formation can lead to enhanced *S. aureus* perseverance on AD skin [223]. However, other research studies have shown that NET levels are not enhanced in AD patients [224]. This highlights that AD is a heterogeneous and complicated disease. Differences in neutrophil-derived antimicrobial peptide expression have been found in AD skin lesions at various stages [225]. Further investigations are needed to figure out whether NETs could influence the expression of antimicrobial peptides in patients with AD [213,224].

#### 5.4.2. Neutrophil Heterogeneity

Patients with AD have considerably higher levels of NLR than healthy controls [226]. NLR in AD patients indicates uncontrolled inflammatory responses and disease severity [227]. Additionally, group 2 innate lymphoid cells (ILC2s) have emerged as critical effector immune cells in triggering allergic reactions in AD. They are crucial producers of type 2 cytokines. Polymorphonuclear myeloid-derived suppressor cells (PMN-MDSCs) as a subpopulation of neutrophils are necessary for ILC2 function to be effectively suppressed. Therefore, increasing PMN-MDSCs might be helpful for controlling ILC2-driven AD [228,229].

In summary, neutrophil-derived molecules and their heterogeneity may contribute to the pathogenesis of AD and exacerbate the disease by various mechanisms. Further research is required to clearly understand the mechanisms underlying the role of neutrophils and their heterogeneity in AD and to develop new therapies specifically targeting these cells.

### 5.5. Rheumatoid Arthritis

Rheumatoid arthritis (RA) is an autoimmune, chronic, and heterogeneous disease. It can be defined by an increasing symmetric joint inflammation that causes bone erosion, cartilage damage, and impairment. It has recently been clear that RA develops from epigenetic, genetic, and environmental triggers, although immunological variables must also play a significant role [230,231]. The pathophysiology of RA is influenced by dysregulation of neutrophil activity. Neutrophils, the most prevalent leukocytes in affected joints, are crucial for the development and persistence of RA [232,233]. Both synovial fluid (SF) and synovial tissue (ST) from RA joints contain a high concentration of activated neutrophils [234]. Activated neutrophils play a role in inflammation and damage to host tissues through degranulation, which occurs either into the SF or directly onto the joint surface. Neutrophils can migrate to the joint. The migration of neutrophils to the joint is a defining feature of inflammatory arthritides5, notably RA. Neutrophils could also serve as a reservoir for autoantigens that instigate the autoimmune processes that underlie this pathogenic condition. A comprehensive grasp of the intricate involvement of neutrophils in RA is imperative for addressing this persistent autoimmune condition.

#### 5.5.1. Neutrophil-Derived Molecules

Higher levels of MPO have been seen within the inflamed synovial membrane of RA patients, which can promote neutrophil recruitment, elevate inflammation, and increase proliferation of synovial fibroblasts [235]. Furthermore, the level of NE is increased in RA joints because it is selective for a variety of substrates, including collagen, elastin, and fibronectin. NE can cause inflammatory responses that contribute to cartilage degeneration by activating proteinase-activated receptors (PARs) [236]. Cat-G is involved in the destruction and degradation of cartilage in RA. The number of neutrophils and the level of IL-6 in SF are directly connected to Cat-G. Patients with RA have elevated Cat-G activity in their SF. In ST, neutrophils express Cat-G to a lesser extent than synovial lining cells [237,238,239]. Moreover, serum α-defensins can serve as helpful markers for estimating the severity of the disease and periods of remission in RA patients. Levels of α-defensins are greater in patients with active RA than in those who are in remission [240]. Neutrophil-derived LTF can act as an endogenous ligand for TLR4 during inflammation of RA synovial fibroblasts (RASFs). LTF can increase mRNA expression levels of cytokines such as IL-6, CCL20, and IL-8 in RASFs activated by TNF-α [241]. The highest levels of salivary LL-37 have been reported in the RA group with chronic periodontitis [242]. Furthermore, Hoffmann et al. [243] have examined expression levels of LL-37 in RA patients and rats in comparison with healthy joints. According to their findings, synovial membranes and joints of RA patients as well as RA rats all showed elevated levels of LL-37. Patients with RA had considerably higher NGAL concentrations in their SF than those with osteoarthritis (OA). NGAL can eradicate the proliferative action of epidermal growth factor (EGF) and fibroblast growth factor (FGF)-2 [244]. Moreover, MMP-9 can increase joint destruction by promoting the survival, invasion, and release of inflammatory cytokines by synovial fibroblasts of RA [245]. MMP-2 and MMP-9 have considerably higher levels in plasma samples of RA patients with vasculitis than from those without vasculitis [246]. The ability of neutrophils to generate ROS is greater in blood and SF samples of RA patients than healthy controls. Increased production of ROS can result in oxidative stress, which might cause tissue damage and ultimately make RA more chronic [247]. Likewise, in murine arthritis, enhanced production of LTB4 and IL-1β allows RA neutrophils to trigger their recruitment [248]. LTB4 levels are increased in serum, SF, and ST samples of RA patients in comparison with healthy controls [249]. These findings imply that LTB4 and its receptor BLT1 are probably involved in the inflammation seen in inflammatory arthritis [250]. Numerous investigations have shown that RA patients exhibit significantly higher concentrations of calprotectin in both SF and serum samples compared to healthy controls [251]. According to previous reports, the generation of pro-inflammatory molecules, cell migration, apoptosis, and cell differentiation are all linked to calprotectin [252,253]. In RA patients, neutrophils can produce large amounts of several inflammatory cytokines (TNF-α, IL-1β, IL-6, IL-8, IL-17β, IL-20, IL-22) that mediate increased neutrophil autophagy [254]. Transcriptomics research has revealed that RA SF neutrophils express a variety of chemokines (CXCL1, CXCL2, and CXCL8) at higher levels as compared to healthy control neutrophils. These chemokines are essential for controlling the inflammation response in the joint [234].

Moreover, patients with RA have PAD2 and PAD4 in their ST and SF [255]. In addition, Spengler et al. [256] have reported that neutrophils that undergo NETosis in the joints of RA patients can generate active PAD, which can help provide citrullinated autoantigens. RA autoantibodies and other proinflammatory cytokines can cause neutrophils to produce NETs [257]. In blood and synovium samples of RA patients, increased NET formation has been reported [258,259]. The role of these compounds allows significant infiltration of neutrophils into the synovium. Furthermore, they can help prolong the survival of cells within the joint by improving the recruitment and migration ability of neutrophils.

#### 5.5.2. Neutrophil Heterogeneity

NLR in the blood may indicate the level of neutrophil-associated inflammation within the synovial membrane of RA patients. NLR is believed to be able to predict responses to DMARD therapy in the progression of RA [260]. High levels of LDGs have been found in the blood of RA patients [261]. Although treatment has little effect on LDG levels in RA blood, LDG levels are correlated with markers of disease activity [261,262]. G-MDSCs can prevent autologous T lymphocytes from becoming activated and proliferating by producing ROS at the immunological synapse [263]. Blood and SF samples of RA patients contain G-MDSCs that can prevent T-cell proliferation [264,265]. In mouse models of arthritis, G-MDSCs can increase the amount of Treg cells while inhibiting T cell proliferation and differentiation into Th1 and Th17 cells [266,267,268]. The heterogeneity of neutrophils in RA demonstrates the complicated nature of this disease.

### 5.6. Systemic Lupus Erythematosus

Systemic lupus erythematosus (SLE) is a chronic, complex, prototypical autoimmune, and worldwide disease. SLE is influenced by environmental, endocrine, immune, and genetic predispositions. Moreover, the body’s immune system overproduces autoantibodies, which results in widespread tissue damage and inflammation. Increased autoantibodies and accumulation of immune complexes are major hallmarks of SLE patients [269,270,271]. Up to this point, the majority of pathology research has concentrated on abnormalities of adaptive immune responses. However, the pathophysiology of SLE has currently received significant interest in innate immune responses, which operate earlier, before adaptive immune responses. Particularly, neutrophils exhibit multiple aspects of dysregulation. Neutrophils exhibit epigenetic alterations and genomic modifications. Furthermore, neutrophils play a major role in the development of SLE by encouraging exposure to self-antigens and generation of autoantibodies. Recently, it has been found that neutrophil ferroptosis has a significant pathogenic impact on SLE. Moreover, interferonopathies, which are defined by excessive type 1 IFN production due to genetic mutation, have a crucial role in SLE. This dysregulated type 1 IFN production promotes the generation of autoantibodies, which is an important aspect of SLE [272]. Dysregulated innate immunity is sufficient to upset the balance of immunological tolerance [273,274,275]. Overall, neutrophils play a complicated role in SLE.

#### 5.6.1. Neutrophil-Derived Molecules

MPO plays an integral role in the SLE inflammatory process. SLE patients have higher levels of MPO in their plasma than healthy controls [276,277,278]. Levels of NE produced by resting neutrophils from SLE and lupus nephritis (LN) are lower than controls. Interestingly, phorbol-12-myristate-13-acetate stimulation can dramatically enhance the release of NE in patients [279]. Cat-G is the main antigen for ANCAs in SLE. Cat-G-ANCA is significantly higher in the sera of active SLE patients than in inactive patients and normal controls. It is rapidly decreased after treatment with corticosteroid drugs [280]. However, it is necessary to clarify the exact function of Cat-G in SLE. According to previous studies, the concentration of serum α-defensins is a reliable biomarker of LN [281]. It positively correlates with disease activity. It has been demonstrated that anti-ribonucleoprotein antibodies can activate neutrophils in SLE [282]. Neutrophil proteins, including LL37 and HMGB1, are also released to a greater extent in SLE patients than in controls [282]. For NGAL, elevated levels in urine are particularly higher in patients with neuropsychiatric SLE and active LN [283,284]. MMP-9 has multiple roles in the development of tissue destruction and inflammation. Several authors have found that serum levels of MMP-9 are higher in SLE patients than in healthy controls [285]. In contrast, certain researchers have not found a significant difference in MMP-9 between those with SLE and those without SLE [286,287]. A recent study [288] has revealed that patients with SLE show hypomethylation in the MMP-9 promoter region. The same study also found a relationship between MMP-9 methylation level and renal involvement, suggesting that MMP-9 methylation level could be used as a diagnostic biomarker for SLE. ROS production is impaired in neutrophils of patients with SLE. Insufficient production of ROS can elevate IFN-α, which is important for preventing tolerance and triggering the pathogenesis of SLE [289,290]. Serum levels of LTB4 are markedly higher in patients with SLE than in healthy controls [291]. Calprotectin has also recently been linked to the pathophysiology of SLE. Compared to controls, SLE patients have significantly higher serum levels of calprotectin [292,293]. The main way that neutrophils contribute to the onset of SLE is by releasing type I interferons (IFN-I) along with additional pro-inflammatory cytokines (IL-1β, IL-18, TNF-α) and inducing systemic tissue damage [115,275]. Chemokines including CXCL1, CXCL2, CXCL3, CXCL5, and CXCL8 are also linked to neutrophil chemoattractants. These chemokines have been found in patients with SLE and LN [17,294].

PAD4 significantly regulates TLR7-mediated autoimmunity in SLE. PAD4 could have a pathogenic function in SLE because of its role in NET formation [295]. The production of autoantigens in SLE is associated with NETs and a related death pathway, NETosis. Additional research has demonstrated that NETs can stimulate plasmacytoid dendritic cells capable of producing IFN-I, an important cytokine in the pathophysiology of SLE [282,296,297]. Therefore, it can be deduced that a lack of balance between NET development and removal in SLE patients may significantly contribute to disease deterioration.

#### 5.6.2. Neutrophil Heterogeneity

NLR is higher in SLE patients. It is associated with significant immunopathological processes such as IFN-I production and neutrophil activation [298]. SLE patients possess LDGs that have an increased ability to produce proinflammatory cytokines such as IFN-I and TNF, which can dramatically increase the risk of vascular injury. Thus, LDGs might be important in the pathophysiology of SLE [299]. G-MDSCs are essential for triggering IFN-I signaling in B cells, which has a substantial impact on the pathogenesis of SLE [300]. CD177+ neutrophils also have a role in SLE. Blood samples from SLE patients are substantially more likely to show co-expression of mature PR3 and CD177+ neutrophils than those from healthy controls [301]. Recent studies have shown that the pathophysiology of SLE might be influenced by neutrophil heterogeneity [302]. More research is needed to determine how neutrophil heterogeneity plays a role in the pathophysiology of SLE. Overall, these findings indicate that neutrophil-secreted substances and a diverse neutrophil population contribute to the development and maintenance of an autoimmune response in SLE.

## 6. Therapeutic Interventions Targeting Neutrophil-Derived Molecules in Inflammatory Diseases

According to findings outlined in previous sections, neutrophils have a vital role in inflammatory conditions. Uncontrolled or inappropriate activity of neutrophils could be a factor in tissue damage in inflammatory conditions and other diseases. Moreover, excessive release of neutrophil-derived molecules and neutrophil heterogeneity can promote inflammatory responses in neutrophil-mediated diseases. In such circumstances, therapeutic approaches to target neutrophils would be able to suppress neutrophil activity by inhibiting excessive secretion of neutrophil-derived molecules. As listed in Table 1, specific targets and inhibitors can be used to modulate neutrophil function in disease models such as MS, AD, IBD, BD, RA, and SLE. Nevertheless, future studies are still needed to reduce treatment side effects and improve function.

## 7. Conclusions

Neutrophils are considered vital cells of our innate immunity with a main function in host defense. Besides being a significant player in innate immunity, a growing body of research suggests that neutrophils have a variety of functions in many infectious and inflammatory diseases. The main way neutrophils aggravate disease is by producing neutrophil-derived molecules and NETs. Neutrophil-derived molecules, ROS, and NETs have a double-edged effect on the severity of many inflammatory diseases. Furthermore, neutrophils demonstrate a wide variety of phenotypes. Recent evidence reveals the presence of specific neutrophil heterogeneity in inflammatory diseases. Nonetheless, their properties, prevalence, and pathological potential require further investigation. Although there have been immense advances in our awareness of the functions of neutrophils in diseases, there is still more to learn regarding the mechanisms behind the migration of neutrophil-derived molecules to different tissues and the function of neutrophil heterogeneity in pathological conditions.

Moreover, several researchers have concentrated their attempts on targeting neutrophil activity as a potential treatment. Many inhibitors and compounds have been recognized as therapeutics by targeting neutrophils over the years. Despite that, more studies should focus on limiting negative effects of inhibitors and substances used for preventing massive release of neutrophil-derived molecules and NET formation. Furthermore, reducing or blocking neutrophil heterogeneity needs additional knowledge of their origin and function in inflammatory diseases and immunological defense. Altogether, understanding neutrophil activity, neutrophil-mediated pathogenesis, neutrophil-derived molecules, and neutrophil heterogeneity is imperative. These considerations, by particularly targeting pathogenic neutrophils without adversely affecting immunity, may therefore be crucial in the development of novel therapies for inflammatory diseases.

## Figures and Tables

**Figure 1 cells-12-02621-f001:**
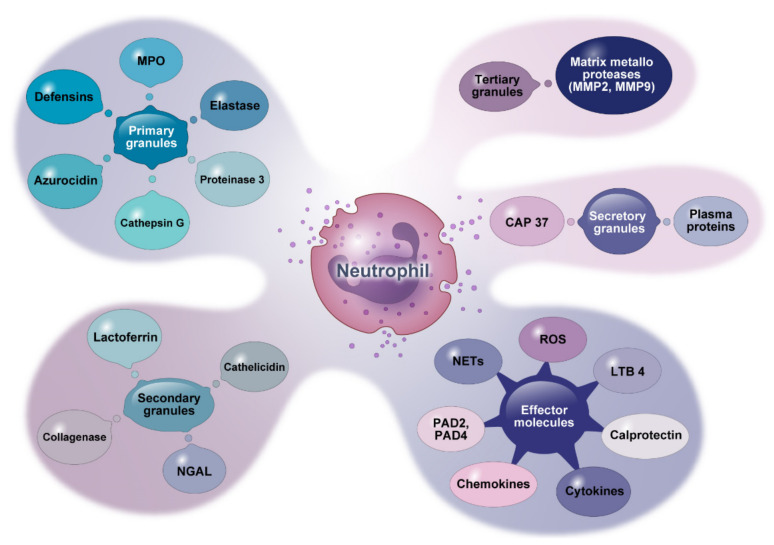
Granule contents and other molecules secreted by neutrophils. This figure shows effective mediators derived from neutrophil granules and other effector molecules they release. MPO: myeloperoxidase. NGAL: neutrophil gelatinase-associated lipocalin. CAP: cationic antimicrobial protein. LTB4: leukotriene B4. PAD: peptidyl arginine deiminases.

**Figure 2 cells-12-02621-f002:**
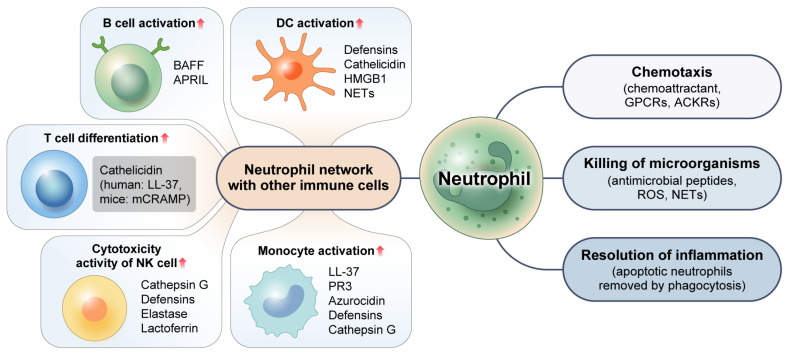
Multifaceted function of neutrophils. Neutrophils eradicate intracellular pathogens through their chemotaxis activity and the release of antimicrobial peptides and ROS. By efficiently engulfing and removing apoptotic neutrophils via the process of phagocytosis, neutrophils can facilitate the resolution of inflammation. Neutrophils can also promote immune responses via complex interactions with other immune cells and use various mechanisms to control activities of other immune cells. GPCRs: G protein-coupled receptors. ACKRs: atypical chemokine receptors. ROS: reactive oxygen species. NETs: neutrophil extracellular traps. HMGB1: high-mobility group box 1. BAFF: B-cell activating factor. APRIL: a proliferation-inducing ligand. PR3: proteinase 3.

**Figure 3 cells-12-02621-f003:**
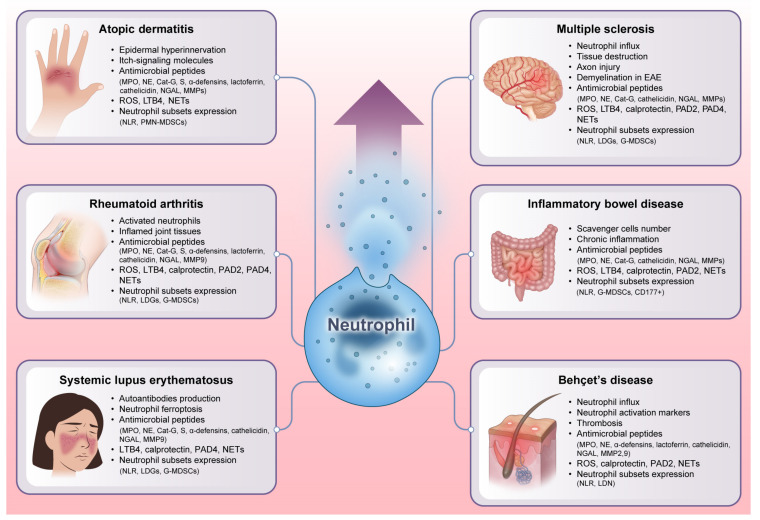
Unraveling the explosive role of neutrophils in inflammatory diseases. Neutrophils exert a profound impact on the development and maintenance of inflammation by secreting antimicrobial peptides and presenting diverse heterogeneity. As immunological orchestrators, neutrophils can also release many other powerful chemicals such as cytokines, chemokines, ROS, and NETs that can intensify the inflammatory cascade. Additionally, the variability of neutrophil subsets, each of which has unique functional characteristics and phenotypes, contributes to the variety of symptoms and severity seen in different inflammatory diseases. All of these factors emphasize that neutrophils can act as vital players in disease pathogenesis. EAE: experimental autoimmune encephalomyelitis. NLR: neutrophil-lymphocyte ratio. LDGs: low-density granulocytes. MDSCs: myeloid-derived suppressor cells. CD: a cluster of differentiation.

**Table 1 cells-12-02621-t001:** Potential or ongoing therapeutics for inhibiting neutrophil-derived molecules.

Target Molecules	Diseases	Compounds/Inhibitors	Administration	Reported Effects	References
MPO	MS	N-acetyl lysyltyrosylcysteine amide (KYC), 4-aminobenzoic acid hydrazide (ABAH)	Intraperitoneal (IP) injection into mice	Diminished axonal injury and demyelination in NOD EAE mice.	[303,304]
	AD	KYC	IP injection into mice	Reduced both ear swelling and vascular permeability in the CHS model.	[204]
	IBD	AZD3241	Oral administration to mice	Ameliorated the MPO-associated tissue damage in the experimental colitis.	[305]
	RA	Tetrandrine	IP injection into mice	Anti-inflammatory effect by significantly decreasing MPO level	[306]
MMP-9	MS	D-penicillamine	IP injection into mice	Inhibited the progression of EAE symptoms	[307]
	AD	Ro 31-9790	Tested on skin-wash samples from AD patients	Suppressed MMP activity	[308]
	IBD	Alpha-lipoic acid, RO28-2653	Oral administration to DSS colitis model.	Protected against UC, acute colitis, and systemic damage	[309,310]
	RA	MMP-9 siRNA	In vitro culture of synovial fibroblasts from RA patients	Suppressed viability of RA synovial fibroblast	[245]
	SLE	Chloroquine phosphate	Oral administration to SLE patients	Reduced serum MMP-9 levels	[311]
Cat-G	IBD	Cat-G inhibitor [Ac-Phe-Val-Thr-PhgP (4-guanidine)-(OC6H4-4-S-Me)2]	In vitro analysis of fecal samples from patients with IBD	Reduced Cat-G activity in both UC and CD.	[312]
	RA	α1-antichymotrypsin, phenylmethylsulfonyl fluoride (PMSF)	In vitro culture of SF from RA patients	Inhibited Cat-G activity	[239]
PAD4	IBD	Cl-amidine	Oral administration to mice	Alleviated clinical colitis and tissue inflammation	[313]
	BD	Cl-amidine	In vitro culture of neutrophils isolated from BD patients	Reduced NETosis	[166]
	RA	JBI-589	Oral administration to mice	Reduced the severity of arthritis	[12]
	SLE	Cl-amidine	Subcutaneous injection to murine lupus model	Inhibited NETs formation and improved thrombosis risk	[314]
NETs	IBD	anti-citrullinated protein antibody (tACPA)	IP injection into mice	Reduced inflammation in colon tissues.	[315]
	BD	Colchicine, Dexamethasone, Apremilast	In vitro culture with neutrophils of patients; oral administration to patients	Inhibited the release of NETs and NETosis	[166,316,317]
	RA	Tocilizumab, polydatin, triptolide, Anti-TNF-α Ab, anti-IL-6 Ab	Administered to mice models and human patients	Reduced NETosis	[258,318,319,320]
	SLE	Rituximab with belimumab, Vitamin D	Administered to SLE patients	Decreased SLE symptoms	[321,322]
ROS	MS	Apocynin (NADPH oxidase inhibitor)	Oral administration to EAE mouse model	Reduced ROS production	[323]
	IBD	Telmisartan (TLM), VAS2870	Oral and IP treatment of colitis rats and mice, respectively.	Reduced ROS production	[324,325]
	BD	N-Acetyl Cysteine (NAC), Apremilast	In vitro culture with neutrophils of patients; oral administration to patients	Inhibited ROS and NETs production	[166,317]
	RA	Mitochondrial division inhibitor 1 (Mdivi-1)	In vitro culture with fibroblast-like synoviocytes from RA patients	Inhibited ROS production and severity of collagen-induced arthritis	[326]
	SLE	Sulforaphane (Nrf2 inducer)	IP injection to MRL/lpr female mice	Neutralized ROS production and improved symptoms	[327]
NE	MS	Sivelestat sodium hydrate (SNa)	IP injection to EAE mice	Reduced Th17-induced EAE	[328]
	AD	DSCG-disodium cromoglycate (Ditec)	Aerosol inhalation to AD patients	In vitro, inhibited the elastase activity.	[206]
	IBD	Sivelestat sodium hydrate (SNa)	Subcutaneously injected into mice	Ameliorated colitis with the reduced level of IL-17	[329]
	RA	EL-17	Oral administration to rat models	Alleviated articular pain	[330]

## Data Availability

Not applicable.

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
