# Peer review of "Neutrophils in Inflammatory Diseases: Unraveling the Impact of Their Derived Molecules and Heterogeneity"

_cells, 2023, doi:10.3390/cells12222621_

Round 1

Reviewer 1 Report

Comments and Suggestions for Authors

In this manuscript, Drs. B. Raiz and S. Sohn have submitted a Review article whose focus is to describe the various molecular role(s) neutrophils might play in the pathogenesis of various human diseases followed by an extensive overview of current results from recent clinical studies addressing how intervention therapies directed toward these molecular entities might eliminate/reduce the underlying pathologies. The Review basically introduces the molecular biology of the neutrophil populations, followed by a presentation of several diseases in which neutrophils could play a major role in addition to a listing of current clinical trial results of several drugs.

Overall, the manuscript is quite well-written and the main theme(s) are well-covered.  However, in the opinion of this Reviewer, the overall message gets lost in the extensive detail of the two sets of diseases (infectious vs immune) each defining multiple examples of different pathological conditions. The descriptions generally detract from the manuscripts main theme, i.e., neutrophils, and seems to emphasize more about the diseases per se. In essence, the listed diseases could easily be listed in Tables, thereby shortening the length of the paper, yet permitting more focus on specific neutrophilic activities.  Furthermore, Figure 2 appears to be more appropriate for Section 3, while Figure 3 appears to be a summary slide that either introduces or summarizes Section 5. 

Minor points:

 1)       In the Introductory (page 2), the authors state that the neutrophil is a homogeneous cell population which can develop multiple subtypes, a perfectively good description.  However, within the text, e.g., page 10 and others, the authors state that the neutrophil is not a homologous cell population but a very diverse heterogenous population with multiple biological activities.  These statements require clarification to avoid confusion.

2)      Title #4 (page 5), neutrophil should be plural

3)   What is the purpose of Figure 3?

Reviewer 2 Report

Comments and Suggestions for Authors

To the Authors,

I have reviewed with interest this paper on the role of neutrophils in inflammatory diseases. The paper is detailed and comprehensive, and could be very useful for the readers. The tables and figures are of good quality and the reference list is adequate. However, in some section the paper is quite hard to follow, as contains a high number of repetitions. Here you will find some suggestions to simplify the flow of the paper:

-Please consider to describe the structure of neutrophils and granules in earlier sections of the paper (they are described in paragraph 5), and then describe the role pf neutrophils in infection and inflammation. 

-Avoid repetitions in the text. Some aspects (i.e general sentences on the role of neutrophils, definition of NETS and specific molecules such as calprotectin, MPO, and others) are described more than 4-5 times in the paper, in different paragraphs. I suggest to describe them only once in a definite section, to make easier to read the paragraph on the single diseases

Concerning the content of the text, I have some minor comments:

-Consider to place the role of neutrophils in a  wider pathogenic context for all the diseases with 1-2 sentences (as correctly made for SLE)

-IBD: it is not completely correct to state that IBD is limited to two entities. it is a spectrum of diseases in which the most well recognised entities are CD and UC, but they are not the only condition

-BD: please consider to better highlight the role of neutrophils in fibrinogen changes and thrombosis, since it is one of the main pathogenic hallmarks of the disease

-Neutrophils and viral infection: the ability of virus to accelerate neutrophils apoptosis has a correlation with post-infectious or intra-infectious neutropenia? Consider to discuss briefly this item

-SLE: consider to mention the role of interferonopathies as condition featured by susceptibility to SLE

I hope that the authors could find the comments as helpful and constructive.
